# Emerging Targeted Therapies for HER2 Positive Gastric Cancer That Can Overcome Trastuzumab Resistance

**DOI:** 10.3390/cancers12020400

**Published:** 2020-02-10

**Authors:** Seiichiro Mitani, Hisato Kawakami

**Affiliations:** Department of Medical Oncology, Faculty of Medicine, Kindai University, 377-2 Ohno-higashi, Osaka-sayama, Osaka 589-8511, Japan; seiichiro.mitani@med.kindai.ac.jp

**Keywords:** gastric cancer, HER2, trastuzumab, AGC, resistance

## Abstract

Trastuzumab, a monoclonal antibody to human epidermal growth factor receptor 2 (HER2), has improved survival in patients with HER2-positive advanced gastric or gastroesophageal junction cancer (AGC). The inevitable development of resistance to trastuzumab remains a problem, however, with several treatment strategies that have proven effective in breast cancer having failed to show clinical benefit in AGC. In this review, we summarize the mechanisms underlying resistance to HER2-targeted therapy and outline past and current challenges in the treatment of HER2-positive AGC refractory to trastuzumab. We further describe novel agents such as HER2 antibody–drug conjugates that are under development and have shown promising antitumor activity in early studies.

## 1. Trastuzumab as a First-Line Therapy for Human Epidermal Growth Factor Receptor 2 (HER2)-Positive Advanced Gastric or Gastroesophageal Junction Cancer (AGC)

HER2, also known as ERBB2, belongs to the ERBB family of proteins that also includes the epidermal growth factor receptor (EGFR or HER1), HER3, and HER4. Trastuzumab is a humanized monoclonal antibody that binds HER2 specifically and thereby inhibits its homodimerization and phosphorylation, resulting in inhibition of the proliferation of HER2-overexpressing tumor cells [1].

### 1.1. ToGA Study

After its approval for the treatment of breast cancer, a global phase III trial, ToGA, was conducted for trastuzumab in AGC patients (Table 1) [2]. Enrolled patients were randomly assigned to receive trastuzumab in combination with chemotherapy (cisplatin with a fluoropyrimidine) or chemotherapy alone. Among 3830 screened patients, 810 were diagnosed with HER2-positive cancer, 594 were randomized, and 584 received study treatment. The overall survival (OS), the primary end point of the study, was found to be significantly longer for trastuzumab plus chemotherapy than for chemotherapy alone, with a median OS of 13.8 versus 11.1 months, yielding a hazard ratio (HR) of 0.74 with a 95% confidence interval (CI) of 0.60 to 0.91 and *p*-value of 0.0046. The median progression-free survival (PFS) was 6.7 months for trastuzumab plus chemotherapy compared with 5.5 months for chemotherapy alone (HR of 0.71, 95% CI of 0.59–0.85; *p* = 0.0002). The overall response rate (ORR) was also significantly greater for trastuzumab plus chemotherapy than for chemotherapy alone: 47% versus 35% (odds ratio of 1.70, 95% CI of 1.22–2.38; *p* = 0.0017).

A preplanned exploratory analysis revealed that patients with a low level of HER2 expression (immunohistochemistry (IHC) score of 0 or 1+ and fluorescence in situ hybridization (FISH)-positive) were less likely to benefit from trastuzumab therapy than those with a high level [2]. On the basis of these results, trastuzumab was approved for AGC with a high HER2 expression level, and trastuzumab-containing regimens are now a standard option for the first-line treatment of such patients, who accounted for 7% to 17% of all individuals with gastric cancer [3,4,5].

### 1.2. Derivatives of the ToGA Regimen in the First-Line Setting

The ToGA trial adopted a regimen of cisplatin combined with either 5-fluorouracil (5-FU) or capecitabine, whereas subsequent prospective studies found similar treatment outcomes with regimens containing oxaliplatin or tegafur–gimeracil–oteracil (S-1). In a single-arm, nonrandomized phase II trial (HER2-based strategy in stomach cancer (HERBIS)–1) performed in Japan [6], trastuzumab in combination with S-1 plus cisplatin yielded a confirmed ORR of 68%, with a median OS and a median PFS of 16.0 and 7.8 months, respectively, in HER2-positive AGC patients with measurable lesions, with these results being similar to those of the ToGA trial [2]. Similar efficacy was also apparent in AGC patients without measurable lesions (HERBIS-1B study) [7]. Three phase II studies that assessed the combination of trastuzumab with capecitabine plus oxaliplatin reported a median OS, a median PFS, and an ORR of 13.8 to 21.0 months, 7.1 to 9.8 months, and 46.7% to 67.3%, respectively [8,9,10]. Trastuzumab in combination with S-1 plus oxaliplatin was also shown to provide a similar treatment outcome in a phase II study, with a median OS, a median PFS, and an ORR of 18.1 months, 8.8 months, and 70.7%, respectively [11]. A meta-analysis of data from these trials revealed that S-1 or oxaliplatin can substitute effectively for capecitabine or 5-FU or for cisplatin, respectively [12].

Immune checkpoint inhibitors such as antibodies to programmed cell death-1 (PD-1) have recently revolutionized treatment strategies for advanced cancer. Given that trastuzumab was found to stimulate T cell responses [13], the combination of trastuzumab-containing regimens with antibodies to PD-1 is receiving attention. A phase II study including 37 patients with HER2-positive AGC treated in the first-line setting with capecitabine, oxaliplatin, and trastuzumab in combination with the anti-PD-1 antibody pembrolizumab reported an ORR of 83%, with a median PFS of 11.4 months and a median OS of not reached [14]. A placebo-controlled, randomized phase III trial (KEYNOTE-811, NCT03615326) is currently ongoing in an attempt to confirm these promising findings.

## 2. Failure of HER2-Targeted Therapy in AGC

For breast cancer, the development of HER2-targeted therapy has been successful [1,15,16,17,18,19,20]. In patients with HER2-positive breast cancer refractory to trastuzumab-based therapy, continuation of trastuzumab in the second-line setting has been shown to prolong survival, with such trastuzumab beyond progression (TBP) being an established strategy for this cancer [15,16]. In addition, agents other than trastuzumab have been found to be effective for HER2-positive breast cancer refractory to trastuzumab. Lapatinib, an oral small-molecule tyrosine kinase inhibitor (TKI) of both HER2 and EGFR, thus confers a significant survival benefit in HER2-positive breast cancer patients when combined with capecitabine or paclitaxel [17,18]. Trastuzumab emtansine (T-DM1) is an antibody–drug conjugate comprised of trastuzumab joined by a stable linker to the microtubule inhibitor emtansine (DM1). T-DM1 is considered a standard care for patients with HER2-positive breast cancer on the basis of the finding that it significantly improves survival outcome in such patients pretreated with trastuzumab [19]. Pertuzumab, a recombinant monoclonal antibody to HER2 that binds to a different domain of the receptor compared with that targeted by trastuzumab, was also shown to prolong survival in HER2-positive breast cancer when added to trastuzumab plus chemotherapy [20].

Numerous clinical trials including phase III studies have been performed for HER2-positive AGC in an attempt to establish new options for HER2-targeted therapy. However, no positive data have been obtained to date.

### 2.1. Trastuzumab in the Second-Line Setting (beyond Progression)

A randomized phase II study (T-ACT, WJOG7112G) examined the efficacy of TBP in combination with paclitaxel, the standard of care in the second-line setting for AGC patients, who progressed during the first-line treatment with a trastuzumab-containing regimen (Table 1) [21]. A total of 89 patients with HER2-positive AGC, who failed first-line therapy with trastuzumab plus a fluoropyrimidine and platinum agent, were randomly assigned to receive paclitaxel plus trastuzumab or paclitaxel alone. No significant difference in PFS or OS between the two arms was detected with the trial, thus failing to demonstrate a benefit for the TBP strategy.

### 2.2. T-DM1 and Pertuzumab

The GATSBY trial was an open-label, adaptive phase II/III study that compared T-DM1 with the physician’s choice of taxane regimen for HER2-positive AGC in the second-line setting (Table 1) [22]. This trial found that T-DM1 treatment did not improve the primary end point of OS or a secondary end point of PFS. A phase III study (JACOB) that evaluated the effect of the addition of pertuzumab to the standard ToGA regimen in HER2-positive AGC also failed to achieve its primary end point of OS, despite significant improvements in PFS and ORR (Table 1) [23].

### 2.3. Lapatinib

The survival benefit of lapatinib for HER2-positive AGC has also been evaluated in a couple of phase III trials. The TRIO-013/LOGiC trial examined the effect of the addition of lapatinib to the combination of capecitabine and oxaliplatin for the first-line treatment of HER2-positive AGC (Table 1) [24]. This study found that lapatinib addition to chemotherapy did not result in a significant improvement in OS, although the PFS and the ORR (53% versus 39%, *p* = 0.003) both favored the lapatinib arm. In the TyTAN trial, which examined paclitaxel with or without lapatinib in the second-line setting, no OS benefit was apparent for lapatinib (Table 1) [25]. No significant improvement in PFS or time to progression was observed, despite significant increases in ORR for all enrolled patients as well as for those with an IHC score of 3+ for HER2.

In summary, although HER2-targeted therapy has been found to improve short-term outcome in HER2-positive AGC, the success of such therapy achieved in HER2-positive breast cancer has not been reproduced in patients with AGC, highlighting the importance of understanding the mechanisms of resistance to HER2-targeted therapy in AGC.

## 3. Mechanisms of Resistance to HER2-Targeted Therapy

Several potential mechanisms of resistance to HER2-targeted therapy in breast cancer have been identified. These mechanisms are as follows: (1) hindering of the access of trastuzumab to HER2 by expression of an extracellular domain-truncated form of HER2 (p95 HER2) or overexpression of MUC4; (2) alternative signaling from the insulin-like growth factor-1 receptor, other HER family members, or mesenchymal–epithelial transition (MET); (3) aberrant downstream signaling caused by loss of PTEN, *PIK3CA* mutation, or downregulation of the cyclin-dependent kinase inhibitor p27; and (4) Fc gamma receptor 3A gene polymorphisms [26]. HER2-positive AGC has been found to share some of these mechanisms with breast cancer, but also manifests specific mechanisms of resistance to trastuzumab.

### 3.1. Tumor Heterogeneity in HER2 Positivity

Gastric cancer is a highly heterogeneous malignancy. Intratumoral HER2 heterogeneity is more frequent in gastric cancer than in breast cancer, with values ranging widely from 23% to 79% as a result of differences in the definition of heterogeneity among studies [27]. In addition, discrepancies in HER2 status between the primary tumor and metastatic sites have been identified. The GASTHER1 study evaluated HER2 status at metastatic sites of patients with AGC, whose primary tumors were found to be negative for HER2 at an initial screening. This study found that 5.7% of initially HER2-negative patients turned out to have HER2-positive metastatic lesions, with liver metastases being associated with the highest frequency of discordance (17.2%) [28]. A retrospective analysis that investigated differences in HER2 status between the primary tumor and metastatic lymph nodes or other metastatic sites also found that ~10% of cases developed discrepancies [29]. Such tumor heterogeneity increased the risk of a false positive result on HER2 testing, potentially leading to a reduced survival benefit for HER2-tageted therapy in clinical trials. Two retrospective studies in Japan indeed detected a poorer outcome of trastuzumab-based therapy in AGC patients with heterogeneity of HER2 expression than in those with homogeneity of such expression [30,31]. A recent study analyzed gene alterations by next-generation sequencing in 50 patients with HER2-postive metastatic esophagogastric cancer, who received first-line trastuzumab therapy [32]. Four patients, whose HER2 status was positive by FISH or IHC, were shown to be negative for *HER2* amplification by next-generation sequencing and progressed rapidly on trastuzumab therapy. Such discordance between FISH/IHC and next-generation sequencing may reflect intratumoral heterogeneity and lead to a poor treatment outcome.

### 3.2. Loss of HER2 Protein Expression

Studies that evaluated changes in HER2 protein expression in AGC patients receiving trastuzumab-containing regimens by comparison of matched pre- and post-treatment samples have demonstrated a loss of HER2 [33,34]. In the aforementioned T-ACT study [21], given that collection of new tumor biopsy samples at the time of study enrollment (after progression on prior trastuzumab therapy) was not mandatory, reassessment of HER2 positivity was performed for only 16 out of 44 patients. Nonetheless, 11 of these 16 patients (69%) were found to lose HER2 positivity [21], with such a loss likely contributing to the failure of TBP in this study. In the study by Janjigian et al., the comparison of matched pre- and post-trastuzumab samples revealed loss of *HER2* amplification [32].

### 3.3. Alterations in HER2 Downstream Signaling

The abovementioned study that applied next-generation sequencing also analyzed gene alterations related to trastuzumab resistance [32]. Alterations that affected the receptor tyrosine kinase (RTK)–RAS–phosphatidylinositol 3-kinase (PI3K) signaling pathway were associated with a short time to treatment failure for trastuzumab therapy. This association is consistent with findings for breast cancer showing that aberrant downstream signaling can give rise to resistance to HER2-targeted therapy. Heregulin serves as a ligand of HER3 and triggers HER2–HER3 heterodimerization and activation of PI3K–AKT signaling [35,36,37]. We previously found that high levels of heregulin in tumor specimens were associated with resistance to trastuzumab in both HER2-positive breast cancer and AGC [38].

### 3.4. Bypass Pathways

The Cancer Genome Atlas classification [39] identifies four molecular subtypes of gastric cancer: tumors with microsatellite instability, tumors positive for Epstein–Barr virus, genomically stable tumors, and tumors with chromosomal instability (CIN). The CIN subtype, which is the most common subtype in gastric cancer, is characterized by marked aneuploidy. CIN is also linked to amplification in oncogene RTK signaling pathways, including EGFR and MET in addition to HER2. Surgically resected gastric tumors were found to manifest a significant association between HER2 protein expression and MET protein expression [40]. A study of *MET*-amplified AGC revealed frequent co-amplification of RTK genes, with 40% to 50% of cases showing co-amplification of either *HER2* or *EGFR*. Such patients failed to respond to HER2-targeted therapy, whereas combined MET and HER2 inhibition was associated with a marked clinical response in one patient [41]. 

Another oncogene co-amplified with *HER2* in AGC is *CCNE1*, which encodes the cell cycle regulator cyclin E1. The application of next-generation sequencing in a phase II study of lapatinib with capecitabine and oxaliplatin in 32 chemotherapy-naive patients with HER2-positive AGC revealed a high frequency (40%) of *CCNE1* co-amplification [42]. Nonresponders to lapatinib treatment were more likely to manifest *CCNE1* co-amplification than responders, suggesting that *CCNE1* amplification is a negative predictive factor. A retrospective study also found that a higher level of copy number variation for *CCNE1* correlated with a shorter survival time in patients with HER2-positive AGC treated with trastuzumab [43]. Of note, co-amplification of *CCNE1* was found to be more strongly associated with HER2-positive AGC than with HER2-positive breast cancer [44].

Together, these observations suggested that the development of new HER2-targeted therapeutic approaches should take into account challenges posed by concomitant gene alterations in addition to HER2 heterogeneity and loss of HER2 protein expression.

## 4. Novel HER2-Targeted Approaches

### 4.1. ZW25

ZW25 is a bispecific antibody directed toward two different HER2 epitopes, extracellular domain 4 (ECD4) and ECD2 [45], which are the binding sites of trastuzumab and pertuzumab, respectively. The preclinical analysis has revealed that ZW25 manifests antitumor activity over a range of HER2 expression levels and inhibits HER2 signaling more potently than either trastuzumab or pertuzumab. In a phase I basket trial, single-agent ZW25 showed encouraging efficacy in heavily pretreated patients with HER2-positive gastroesophageal cancer, with an ORR of 44% and a disease control rate (DCR) of 56% [46]. Toxicities were manageable with almost all adverse events classified as grade 1 or 2. Only one patient developed toxicities of grade 3 including reversible hypophosphatemia, arthralgia, and fatigue, and there were no treatment-related deaths. On the basis of these promising results, ZW25 received a fast track designation by the U.S. Food and Drug Administration (FDA), and a trial evaluating ZW25 plus chemotherapy for patients with HER2-positive tumors (NCT02892123) is ongoing.

### 4.2. Margetuximab

Margetuximab is a monoclonal antibody that binds to the same epitope of HER2 (ECD4) as trastuzumab does [47]. Although the affinity of margetuximab for HER2 is similar to that of trastuzumab, the modification of the Fc domain of margetuximab resulted in enhancement of antibody-dependent cell-mediated cytotoxicity compared with that observed with trastuzumab. A total of 66 patients with HER2-positive tumors, including 20 with gastroesophageal cancer, were enrolled in a first-in-human phase I study of margetuximab [48]. Most patients (45 out of 66, ratio: 68%) underwent at least one previous HER2-targeted therapy in the metastatic setting. Adverse events of grade 3 or 4 included increased blood lipase, a decreased lymphocyte count, increased blood amylase, increased blood alkaline phosphatase, and infusion-related reaction, all of which occurred in <5% of patients. Among 60 patients evaluable for tumor response, seven individuals including one with gastroesophageal cancer showed a confirmed partial response, resulting in an ORR of 12%. For breast cancer, the phase III SOPHIA trial compared margetuximab plus chemotherapy with trastuzumab plus chemotherapy in such heavily pretreated patients [49]. Margetuximab plus chemotherapy demonstrated significant improvements in its primary end point compared with chemotherapy alone (median PFS: 5.8 versus 4.9 months, HR of 0.76, 95% CI of 0.59–0.98; *p* = 0.033). Margetuximab in combination with pembrolizumab is currently under investigation for AGC (NCT02689284). The preliminary results in patients with HER2-positive gastroesophageal adenocarcinoma, who progressed on first-line trastuzumab treatments, have been already reported. Margetuximab in combination with pembrolizumab as a second-line setting showed acceptable toxicities with 18.2% of grade 3 or more treatment-related adverse events such as autoimmune hepatitis. Patients with an IHC score of 3+ were more likely to gain clinical benefit with an ORR of 41%, a median PFS of 5.5 months, and a median OS of not reached [50].

### 4.3. Pan-HER TKIs

Several HER2-targeted TKIs have been evaluated in clinical trials. Whereas lapatinib targets EGFR (HER1) and HER2, recent trials have focused on pan-HER inhibitors, given that studies have suggested that the antitumor efficacy of pan-HER blockade is more promising than that of HER2 blockade alone [51]. 

Afatinib is an oral TKI that irreversibly blocks EGFR, HER2, HER3, and HER4, and it has shown promising preclinical activity against HER2-positive gastrointestinal tumors that are resistant to trastuzumab [52]. A phase II study that evaluated afatinib monotherapy in 20 patients with esophagogastric cancer previously treated with trastuzumab found that afatinib provided moderate therapeutic benefit with an ORR of 10%, and the data suggested that co-amplification of *EGFR* and *HER2* predicted treatment response [53]. 

Neratinib is another pan-HER TKI that binds irreversibly to the active site of the tyrosine kinase domain and blocks signal transduction by EGFR, HER2, and HER4 [54]. Neratinib has been tested against HER2-mutated tumors, and a nonrandomized phase II basket study (SUMMIT, NCT01953926) is currently exploring its efficacy.

Tucatinib is an oral TKI that is highly selective for HER2 and has shown clinical benefit for patients with HER2-positive tumors, especially for those with central nervous system metastasis [55]. Tucatinib was granted fast track designation by the U.S. FDA for the treatment of HER2-positive breast cancer. The HER2CLIMB trial evaluated the addition of tucatinib to trastuzumab and capecitabine in patients with HER2-positive breast cancer previously treated with trastuzumab, pertuzumab, and T-DM1. The one-year PFS rate was 33.1% for the tucatinib-containing regimen and 12.3% for the control regimen (HR of 0.54, 95% CI of 0.42–0.71; *p* < 0.001), and the median PFS rates were 7.8 and 5.6 months, respectively, with the primary end point thus being met [56].

### 4.4. Trastuzumab Deruxtecan (DS-8201a)

Trastuzumab deruxtecan (DS-8201a) is a novel HER2-targeted antibody–drug conjugate. The antibody of trastuzumab deruxtecan was developed with reference to the amino acid sequence of trastuzumab and thus binds to HER2 with a similar affinity. The drug payload is a derivative of the topoisomerase I inhibitor DX-8951 (DXd) and shows a higher potency compared with SN-38, the active metabolite of irinotecan [57]. Furthermore, trastuzumab deruxtecan has a drug-to-antibody ratio of 8, which is higher than that of T-DM1 (3.5). In addition, the novel linker technology provides a stable and efficient linkage between the antibody and drug payload of trastuzumab deruxtecan. In contrast to T-DM1, these unique characteristics of trastuzumab deruxtecan render it effective against tumors with low levels of HER2 expression [58].

In a first-in-human phase I study, patients with advanced breast cancer or AGC received trastuzumab deruxtecan at a dose of 0.8 to 8.0 mg/kg intravenously every three weeks [59]. Common adverse events of grade of ≥3 included myelosuppression as reflected by a decreased lymphocyte count, a decreased neutrophil count, and anemia. Three serious adverse events (febrile neutropenia, intestinal perforation, and cholangitis) occurred in one patient each. However, no dose-limiting toxic effects were encountered. The ORR and DCR were 43% and 91%, respectively, for patients, who underwent multiple lines of standard therapy. A phase II study for heavily pretreated patients with HER2-positive breast cancer revealed promising efficacy for trastuzumab deruxtecan at a dose of 5.4 mg/kg, with an ORR and a DCR of 60.9% and 97.3%, respectively [60]. The high antitumor efficacy was further translated into prolongation of survival, with a median PFS of 16.4 months (95% CI of 12.7 months—not reached).

A dose-escalation and dose-expansion phase I trial was conducted for trastuzumab deruxtecan in patients with AGC [61]. A total of 44 patients pretreated with HER2-positive AGC received at least one dose of trastuzumab deruxtecan (5.4 or 6.4 mg/kg) every three weeks. Although 11 patients (25%) developed serious treatment-emergent adverse events and there were four cases of pneumonitis, almost all adverse events were consistent with the results of a previously reported phase I study [59]. Nineteen patients achieved a confirmed response, resulting in an ORR of 43.2%, which was also similar to the value in the previous study. The recommended dose for a subsequent phase II study was thus set at 6.4 mg/kg.

The DESTINY-Gastric01 study (NCT03329690), a randomized, open-label phase II trial evaluating the efficacy and safety of trastuzumab deruxtecan versus the physician’s choice of therapy, is ongoing in 220 patients with HER2-positive AGC, who progressed during treatment with two or more previous regimens including trastuzumab. Trastuzumab deruxtecan has received the SAKIGAKE designation for the treatment of HER2-positive AGC by the Japanese Ministry of Health, Labor, and Welfare. SAKIGAKE is a system to place innovative medical products, including pharmaceuticals, medical devices, and regenerative medicine products, into clinical use.

## 5. Advantages of Trastuzumab Deruxtecan

Although homogeneity of *HER2* amplification and expression is necessary for the success of conventional HER2-targeted therapy, such homogeneity is less frequent for AGC than for breast cancer and is not necessarily required for the success of therapy with trastuzumab deruxtecan. We thus previously showed that trastuzumab deruxtecan is effective not only against tumor cells positive for HER2 protein but also, in the presence of HER2-positive cells, against those negative for such expression [62] (Figure 1). This “bystander killing effect” is likely due to the internalization of trastuzumab deruxtecan by HER2-positive cells (Figure 1A), the release of DXd into the cytoplasm of these cells (Figure 1B), and the subsequent transfer of the released DXd into adjacent HER2-negative cells (Figure 1C) [63].

Given that most anti-HER2 drugs including trastuzumab target only the HER2 signaling pathway, their efficacy is limited to *HER2*-amplified tumors, which account for ~17% of AGC tumors. On the other hand, the expression of HER2 protein at various levels on the tumor cell surface occurs more frequently. In AGC, the concordance between IHC and FISH for detection of HER2 overexpression is moderate, with a value of 83% for the ToGA trial [2], suggesting that a substantial population of patients classified as HER2-negative by FISH are actually positive for HER2 protein expression but do not benefit from current anti-HER2 therapy. In our preclinical study, with the use of engineered cell lines that expressed HER2 protein at various levels in the absence of *HER2* amplification, we also demonstrated the efficacy of trastuzumab deruxtecan against tumors that express HER2 but are negative for *HER2* amplification [62]. In this case, HER2 may function as a “gate” for the selective passage of DXd, with the antitumor effect being solely due to the cytotoxicity of DXd, not to HER2 signal blockade by trastuzumab. These preclinical findings may be reflected in the clinical setting by the observation that patients with AGC that expressed HER2 at a low level responded to trastuzumab deruxtecan in the phase I study [59], suggesting that the definition of “HER2 positivity” may need to be changed for similar agents. Furthermore, the antitumor effect of DXd is basically independent of the absence or presence of other gene alterations such as those that activate alternative pathways (*MET* or *CCNE1* co-amplification) or those that affect downstream signaling components like RAS or RAF, suggesting that trastuzumab deruxtecan may be able to overcome trastuzumab resistance. In our preclinical study, we found that the antitumor effect of trastuzumab deruxtecan was more rapid for *HER2*-amplified tumors than for those that expressed HER2 without *HER2* amplification, possibly as a result of a difference in tumor dependence on HER2 signaling [62]. These preliminary results await clinical confirmation, however.

Antibodies to HER family members have been developed mostly for breast cancer and gastrointestinal malignancies including AGC and metastatic colorectal cancer (mCRC). Antibodies to EGFR have been shown to confer a survival benefit in the first-, second- or later-line setting for patients with mCRC that is wild type for *RAS*. However, the benefit of retreatment with such antibodies has been unclear [64]. A recent phase II CRICKET study examined the possibility of rechallenge with antibodies to EGFR in the third-line setting for RAS wild-type mCRC [65]. The study recruited patients, who benefited from first-line treatment with anti-EGFR antibody cetuximab before the development of resistance and administration of chemotherapy as a second-line treatment. It was found that 21% of patients responded to irinotecan plus cetuximab in the third-line setting, suggesting that the sensitivity to the anti-EGFR antibody was restored over the period of subsequent therapy. The mechanism underlying this loss of resistance is unknown, but the second-line chemotherapy in the absence of cetuximab might play a key role [66]. To date, most anti-HER2 agents, which failed to show clinical benefit, have been examined in the second-line setting, and little is known about their efficacy in the third- or later-line setting. The efficacy of anti-HER2 agents is basically dependent on expression of the HER2 protein, and such treatment would therefore be expected to be invalid in the second-line setting for tumors that lose such expression as a result of prior trastuzumab treatment. However, as shown in the CRICKET study, second-line chemotherapy without targeting EGFR might induce recovery of sensitivity to anti-EGFR antibodies. In this context, the results of the DESTINY-Gastric01 study will be of interest, given that the study is being conducted in the third-line or later-line setting. Longitudinal assessment of HER2 protein expression should also provide insight into this issue. 

## 6. Conclusions

After the success of the ToGA study, several anti-HER2 drugs failed to show efficacy against HER2-positive AGC, mostly in the second-line setting. Improvement in clinical outcome for HER2-positive AGC patients will require an understanding of the mechanisms underlying trastuzumab resistance. A new generation of HER2-targeted drugs including bispecific antibodies, antibody–drug conjugates, and TKIs has been designed to overcome such mechanisms. Among them, trastuzumab deruxtecan has the advantage of a unique antitumor mechanism including a bystander killing effect. Several clinical trials evaluating novel anti-HER2 approaches are ongoing, but we are still awaiting a breakthrough for the treatment of HER2-positive AGC.

## Figures and Tables

**Figure 1 cancers-12-00400-f001:**
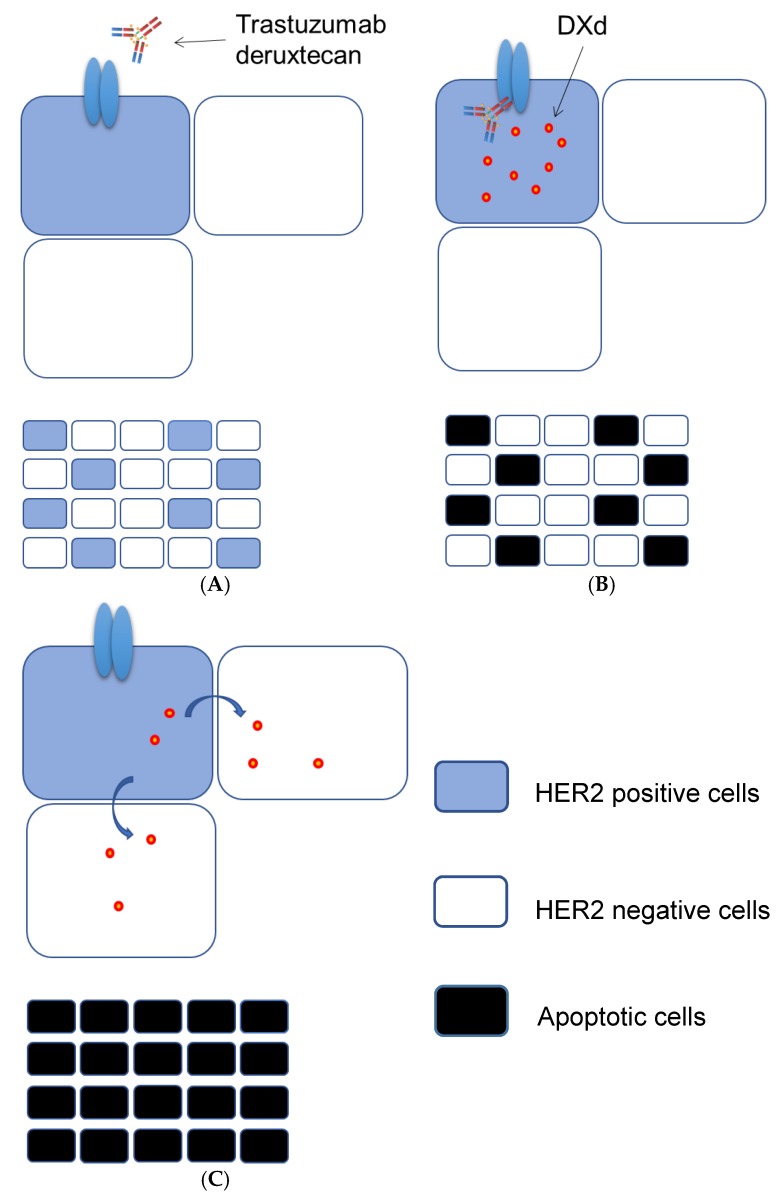
Proposed mechanism for the “bystander killing effect” of trastuzumab deruxtecan. The binding of trastuzumab deruxtecan to HER2 expressed on the surface of HER2-positive tumor cells (**A**) triggers its internalization followed by the release of DXd into the cytoplasm and the induction of apoptosis (**B**). (**C**) DXd is then transferred to and induces apoptosis in neighboring HER2-negative cells.

**Table 1 cancers-12-00400-t001:** Pivotal randomized phase II and III trials of human epidermal growth factor receptor 2 (HER20-targeted agents in HER2-positive advanced gastric or gastroesophageal junction cancer.

Trial	Agent	Line of Therapy	Median PFS (Month)	Median OS (Month)	Result for the Primary End Point
ToGA [2]	Trastuzumab	1st	6.7 versus 5.5 (HR, 0.71; 95% CI, 0.59–0.85; *p* < 0.01)	13.8 versus 11.1 (HR, 0.74; 95% CI, 0.60–0.91; *p* < 0.01)	Positive
LOGiC [24]	Lapatinib	1st	6.0 versus 5.4 (HR, 0.82; 95% CI, 0.68–1.00; *p* = 0.04)	12.2 versus 10.5 (HR, 0.91; 95% CI, 0.73–1.12; *p* = 0.20)	Negative
JACOB [23]	Pertuzumab	1st	8.5 versus 7.0 (HR, 0.73; 95% CI, 0.62–0.86; *p* < 0.01)	17.5 versus 14.2 (HR, 0.84; 95% CI, 0.71–1.00; *p* = 0.06)	Negative
T-ACT [21]	Trastuzumab	2nd	3.2 versus 3.7 (HR, 0.91; 95% CI, 0.67–1.22; *p* = 0.33)	10.0 versus 10.2 (HR, 1.23; 95% CI, 0.76–1.99; *p* = 0.20)	Negative
TyTAN [25]	Lapatinib	2nd	5.5 versus 4.4 (HR, 0.85; 95% CI, 0.63–1.13; *p* = 0.24)	11.0 versus 8.9 (HR, 0.84; 95% CI, 0.64–1.11; *p* = 0.10)	Negative
GATSBY [22]	T-DM1	2nd	2.7 versus 2.9 (HR, 1.13; 95% CI, 0.89–1.43; *p* = 0.31)	7.9 versus 8.6 (HR, 1.15; 95% CI, 0.87–1.51; *p* = 0.86)	Negative

PFS, progression-free survival; OS, overall survival; HR, hazard ratio; CI, confidence interval; T-DM1, trastuzumab emtansine.

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
