# Peer review of "Emerging Targeted Therapies for HER2 Positive Gastric Cancer That Can Overcome Trastuzumab Resistance"

_cancers, 2020, doi:10.3390/cancers12020400_

Round 1

Reviewer 1 Report

This manuscript is comprehensive and well-written. My comments are as below;

In page 1, line 29, the authors wrote ‘promising results had been obtained in phase II studies for advanced gastric or gastroesophageal junction cancer’. However, it would not be true. Please provide a relevant reference.

In page 3, line 83, they wrote ‘Lapatinib, an oral small-molecule tyrosine kinase inhibitor (TKI) of both HER2 and EGFR, thus conferred a significant survival benefit’ Please add ‘when combined with capecitabine or paclitaxel’.

In page 4, line 147, they described a study by Janjigian YY et al. (Cancer Discov. 2018). I think that this part should be moved to the section of ‘Loss of HER2 protein expression’.

In page 5, line 172, they described ‘Bypass pathways’. I think that this part is a little lengthy and redundant. Please shorten this part.

In page 6, line 224, they described ‘Sophia trial in breast cancer’. The results of this study has been presented at the ASCO 2019. And the preliminary results of the combination of margituximab and pembrolizumab in AGC has been also presented at the ASCO-GI 2019.

In page 6, line 252, they described ‘DS-8201a’. The description is too lengthy compared to description of other compounds. Please shorten this part.

In page 9, line 325, they described ‘anti-EGFR antibody’. This part is less relevant, I think. Please remove this part.

Author Response

In page 1, line 29, the authors wrote ‘promising results had been obtained in phase II studies for advanced gastric or gastroesophageal junction cancer’. However, it would not be true. Please provide a relevant reference.

Response: The reviewer’s comment is well taken. We have removed that wrong part (Page 1; line 29–31).

In page 3, line 83, they wrote ‘Lapatinib, an oral small-molecule tyrosine kinase inhibitor (TKI) of both HER2 and EGFR, thus conferred a significant survival benefit’ Please add ‘when combined with capecitabine or paclitaxel’.

Response: Thank you for the reviewer’s careful reading. We have added the sentence as indicated (Page 3; line 85–86).

In page 4, line 147, they described a study by Janjigian YY et al. (Cancer Discov. 2018). I think that this part should be moved to the section of ‘Loss of HER2 protein expression’.

Response: In that article, several potential mechanisms of trastuzumab resistance were discussed. Since discordance between FISH/IHC and next-generation sequencing was reported to be possibly associated with tumor heterogeneity in HER2 positivity, we would like to leave the section of “Tumor heterogeneity in HER2 positivity” untouched. As the reviewer pointed out, however, loss of HER2 expression was also described in the same section. Therefore, we have revised the section titled as “Loss of HER2 protein expression” (Page 4; line 162–163).

In page 5, line 172, they described ‘Bypass pathways’. I think that this part is a little lengthy and redundant. Please shorten this part.

Response: We agree with the reviewer’s suggestion. We therefore have rewritten the text concisely (Page 5; line 177–184).

In page 6, line 224, they described ‘Sophia trial in breast cancer’. The results of this study has been presented at the ASCO 2019. And the preliminary results of the combination of margituximab and pembrolizumab in AGC has been also presented at the ASCO-GI 2019.

Response: We thank the reviewer’s valuable suggestion. We have added the information which were presented ASCO2019 and ASCO-GI2019 (Page 6; line 228–239).

In page 6, line 252, they described ‘DS-8201a’. The description is too lengthy compared to description of other compounds. Please shorten this part.

Response: Trastuzumab deruxtecan (DS-8201a) is definitely closest to approval among promising new anti-HER2 agents listed in our review. Now that trastuzumab deruxtecan was approved for breast cancer, it is quite meaningful to thoroughly review trastuzumab deruxtecan for patients with advanced gastric cancer. In addition, we have been working on preclinical evaluation studies of trastuzumab deruxtecan and therefore are able to discuss in detail. We thank the reviewer’s understanding in this regard.

In page 9, line 325, they described ‘anti-EGFR antibody’. This part is less relevant, I think. Please remove this part.

Response: The development of trastuzumab deruxtecan as third- or later- lines chemotherapy seems optimal, whereas most of anti-HER2 therapy have failed to show its efficacy possibly due to the 2nd line setting. We believe that this point is essential to our review article. In this regard, CRICKET study is a good example as rechallenge of anti-EGFR antibodies was evaluated in the anti-EGFR antibody refractory patients. To clarify our intent, we have supplemented the description regarding anti-EGFR antibodies (Page 9; line 341, Page 9; line 349–355). We thank the reviewer’s understanding.

Reviewer 2 Report

The review article title “Emerging targeted therapies for HER2 positive gastric cancers that can overcome trastuzumab resistance” describes the clinical findings of ToGA trials, with improved overall survival with Trastuzumab in combination with chemotherapeutics as compared to chemotherapeutic alone. It further discussed subsequent studies following ToGA regimen are described well by enlisting trials that show similar response. It states that HER2 targeted therapies works wells for breast cancer but highlighted the failure of HER2 targeted therapy with TBP, T-DM1 and Pertuzumab or, Lapatanib in AGC. Following that the review describe in detail the possible mechanism of resistance of HER2-targeted therapy and emerging future possible targets to overcome the trastuzumab resistance, is very well written.

The mechanism of resistance is described by focusing on the alterations in signaling pathways, heterogeneity, and loss of HER2 protein expression but obstacles that prevent trastuzumab binding, Resistance caused due to cancer stem cells is totally shadowed or not been discussed.

The few comments I suggest to reconsider are as follow:

l.52. Heading “Offspring of the ToGA regimen in the first-line setting” I am not sure if medical terminology is appropriate for such heading. Additionally it creates confusion whether the results of ToGA studies are enlisted or subsequent perspective studies are discussed. The suggested heading can be consider as “Studies mimicking ToGA regimen in the first line setting” or “Descendant studies following ToGA regimen in the first line setting” else it can be put as “trastuzumab in combination with Chemotherapeutic in the first line setting”. 

l.129. and 186. Acronyms details are missing Like MET, HER2 etc. Overall and mentioned lines containing gene symbols can be italicized.   

l.203-205. Corresponding reference should be added for mentioned preclinical studies.

Author Response

52. Heading “Offspring of the ToGA regimen in the first-line setting” I am not sure if medical terminology is appropriate for such heading. Additionally it creates confusion whether the results of ToGA studies are enlisted or subsequent perspective studies are discussed. The suggested heading can be consider as “Studies mimicking ToGA regimen in the first line setting” or “Descendant studies following ToGA regimen in the first line setting” else it can be put as “trastuzumab in combination with Chemotherapeutic in the first line setting”.

Response: In response to the reviewer’s suggestion, we have changed “Offspring of the ToGA regimen” into “Derivatives of ToGA regimen”. If this still causes confusion, we would like to use “Descendant studies following ToGA regimen in the first-line setting”. (Page 2; lines 53).

129. and 186. Acronyms details are missing Like MET, HER2 etc. Overall and mentioned lines containing gene symbols can be italicized

Response: In response to reviewer’s comment, we have spelled out these words (Page 4; line 130). Additionally, we have italicized gene symbols.

203-205. Corresponding reference should be added for mentioned preclinical studies

Response: We thank the reviewer’s careful reading. Reference number is “45”, which is the same as the one listed above.